# OpenReview forum: "Prompt Space Optimizing Few-shot Reasoning Success with Large Language Models"
_ICLR.cc/2024/Conference — ICLR 2024 Conference Withdrawn Submission_

### Official Review · Reviewer_YN58 · 2023-10-30

**Soundness:** 2 fair
**Presentation:** 3 good
**Contribution:** 3 good
**Rating:** 5
**Confidence:** 4

**Summary:**

The paper addresses the challenge of prompt engineering by creating a space for all potential prompts and identifying foundational questions through PCA. The key strength is providing a mathematical framework for prompt selection. The proposed technique outperforms both traditional manual prompts and previous automated selection approaches.

**Strengths:**

1. The paper is easy to read, and generally well written.
2. The idea is intuitive yet effective. By employing principal component analysis within the prompt exemplar space, the authors identify the most informative prompts. This method's effectiveness has been substantiated across ten datasets.
3. Supplementary experiments provide a detailed demonstration of the impact certain experimental settings have on the results.

**Weaknesses:**

1. The PROMPT SPACE exhibits certain instabilities. The choice of encoding model and hyperparameter settings can profoundly impact the model's performance. I hope the authors can provide a more thorough analysis in this regard. Specific concerns are detailed in the questions section.
2. Both the method in this paper and Auto-CoT aim to identify the most representative samples. While experimental results suggest that  PROMPT SPACE is superior, I believe the paper should provide a clearer analysis of why matrix decomposition in the PROMPT SPACE is more effective, compared to clustering.

**Questions:**

1. Why does MiniLM perform the best? What factors might contribute to this? Can we only determine which model to use based on the results from the validation set?
2. If basis exemplars exist, why does increasing their number lead to a decline in performance (Figure 5)? Intuitively, a larger quantity should encompass more information.
3. Some of the numbers in the table are blurry and need to be fixed.

---

### Official Review · Reviewer_46e7 · 2023-10-30

**Soundness:** 2 fair
**Presentation:** 2 fair
**Contribution:** 2 fair
**Rating:** 3
**Confidence:** 4

**Summary:**

The paper presents a new prompt engineering method to select which questions to
include in the prompt. The question set of the task is embedded into a matrix
and it's dimensionality is reduced using PCA, then the questions which are most
similar to the principal components (the basis questions) are selected to be
included into the prompt.

**Strengths:**

PCA has not been used for selecting the questions to include into the prompt as
far as I know.

The method is evaluated on numerous datasets and achieves state-of-the-art
results.

**Weaknesses:**

The main weakness of the method is its sensitivity to the number of principal
components to include, $k$. Figure 5 shows that accuracy varies hugely (by 20 to
30 percent) if we don't choose exactly the right value for $k$. Comparatively
the method achieves modest gains (1 to 3 percent) compared to previous work. The
accuracy drops happen even if we change $k$ by just 1 or 2. The authors state
that they cannot automatically determine the optimal parameter value for a
dataset.

In the Introduction, the authors state that previous methods have "lack of
guidance on finding optimal prompts" in contrast the the presented method. I
don't think this is true as previous methods (including Auto-CoT) are very
similar; the most significant difference is the way the questions to include in
the prompt are selected, which is using PCA in the presented method.

Connected to this, some claims the paper makes are too extensive in my opinion,
like the presented method being "an innovative mathematical solution" which
"aims to develop a deep understanding on how to design CoT prompting".

The algorithm itself is presented as if it would be a novel algorithm inspired
by PCA, but I think it's actually PCA without making the data zero mean. I found
some of the explanations (e.g., linear independence, some parts of PCA, the
proof for using SVD to compute PCA) superfluous.

A weakness of the evaluation is that the authors used gpt-35-turbo instead of
the widely used davinci model. This makes the results very different from
previous work (e.g., the CoT paper reported 46.9% accuracy on GSM8K and 68.9% on
SVAMP with GPT 175B compared to 75.8% and 80.2% in the present paper). Even
though gpt-35-turbo may outperform davinci, it would be good to report those
results too for the sake of comparison to previous work.

Some smaller issues:
- I think that in-context learning is not a prompt engineering technique as
  stated in the abstract.
- In the Introduction, the authors state that CoT prompting demonstrates that
  the reasoning ability of LLMs perfectly matches the scaling laws. I believe
  this should be rephrased as these scaling laws are not mathematically exact to
  be perfectly matched.
- In the first paragraph of 4.2, I think some text is missing from "average of
  3.2% in Table 2".

**Questions:**

Do I understand correctly that the basis questions are the same for a given set
of questions, irrespective of the current question asked? Would it be better to
customize the basis questions for the current question?

---

### Official Review · Reviewer_4To8 · 2023-10-31

**Soundness:** 2 fair
**Presentation:** 1 poor
**Contribution:** 2 fair
**Rating:** 3
**Confidence:** 4

**Summary:**

The paper proposes a method called Prompt Space, that selects k question by SVD from a question set given a task, then these questions (with LLM generated rationales) serve as few-shot exemplars to prompt a language model to give a response. The authors did experiments over 10 reasoning tasks and show their method can improve over a few baselines (zero-shot CoT, few-shot CoT, and Auto-CoT).

**Strengths:**

- The paper presents an interesting idea of finding k most representative questions to serve as few-shot exemplars to improve reasoning in language models.

**Weaknesses:**

The presentation of the method/experiment is very unclear, many of the details are omitted, making it hard to tell if the proposed method is indeed superior compared to existing methods. In addition, many of the design choices (e.g., the question set for each task, number of basis questions $k$, choice of the embedding model) are not well justified and seem to be determined based on the best test set performance (which could mean info-leaking on the test set).

- For doing SVD over a question set: end of page 4, step 1 "Embedding questions", the authors said there are $m$ questions in a task, which they further used to embed and construct a matrix to select the top k basis questions from. Where do those $m$ questions come from? The training set or the test set of each task? and what is $m$ for each task? Note both zero-shot CoT and few-shot CoT are completely unsupervised and do not require a training dataset.

- Optimal number of basis questions: In the beginning of section 4, the authors mentioned " 3. Prompt Space can determine the optimal number of basis questions for significantly improving the performance of
LLMs on each dataset. ". However, in section 4.4, the authors said "there remains a challenge that we cannot automatically determine the optimal number of basis questions for each dataset". Can the optimal number of basis questions be automatically determined or not? Based on Figure 5, there is a big discrepancy on the performance depending on the number of basis questions, how is the final performance determined in Table 1 and 2? Is it based on which k gives the best test performance?

- Which embedding models to use: in Section 4.3, different embedding models give quite different results, for example, if choosing a different embedding model, the proposed method's performance could be worse compared to the baselines. Why choosing the MiniLLM:384 as the embedding model? Is it again based on the test set performance?

**Questions:**

- What is the question set for each task? what is $m$ for each task?

- How is $k$, the optimal number of basis questions chosen?

- Why the miniLLM model is chosen as the embedding model?

---

### Official Review · Reviewer_bdzS · 2023-10-31

**Soundness:** 2 fair
**Presentation:** 2 fair
**Contribution:** 1 poor
**Rating:** 3
**Confidence:** 4

**Summary:**

The paper studies the problem of engineering Chain-of-thought prompts. It extends the Auto-CoT approach by selecting more representative examples. Specifically, it maps questions onto a vector space and uses matrix decomposition to obtain basis vectors that represent the question space, and use zero-CoT to automatically obtain chain-of-thoughts in order to get the full prompts. The paper experiments on multiple reasoning datasets to study the effectiveness of the proposed method.

**Strengths:**

The proposed approach is intuitive and straightforward.


The experiment considers multiple datasets and multiple tasks. The paper also provides some analysis of impact on number of examples and order of examples.


The paper is mostly clear.

**Weaknesses:**

The paper misses strong example selection baselines. The paper mainly extends the Auto-CoT in the example selection part, and compares it to Auto-CoT (where examples are selected from clusters). There are many more advanced (e.g., learning-based methods) example selection strategy like [1][2][3][4][5][6]


[1] Learning To Retrieve Prompts for In-Context Learning. EMNLP 2022.

[2] Dynamic Prompt Learning via Policy Gradient for Semi-structured Mathematical Reasoning. ICLR 2023.

[3] Complexity-Based Prompting for Multi-Step Reasoning. ICLR 2022.

[4] Compositional Exemplars for In-context Learning. ICML 2023

[5] complementary explanations for effective in-context learning. ACL Findings 23.

[6] Automatic Prompt Augmentation and Selection with Chain-of-Thought from Labeled Data. Arxiv 23



---



The paper claims the proposed approach can choose an optimal number of examples. From section 3, I don’t see a very clear and formal description of how the optimal number of examples are selected. Is it achieved using a validation set (which seems to be suggested on 4.4)?  Also, as in 4.4, it suggests the question space of AQUA (a more diverse and complex dataset) is even smaller than quite simple formulaic  datasets like SingleEQ, which seems to invalidate the claim of the paper on that the prompt space method selects questions that represent the space; intuitively, the AQUA should not be less diverse than the more synthetic and narrow datasets like SingleEQ.

**Questions:**

How the optimal number of examples are chosen?